# Accuracy and Interpretation of the Acceleration from an Inertial Measurement Unit When Applied to the Sprint Performance of Track and Field Athletes

**DOI:** 10.3390/s23041761

**Published:** 2023-02-04

**Authors:** Paulo Miranda-Oliveira, Marco Branco, Orlando Fernandes

**Affiliations:** 1Interdisciplinary Research Centre Egas Moniz (CiiEM), Egas Moniz School of Health & Science, 2829-511 Almada, Portugal; 2School of Technology and Management (ESTG), Polytechnic of Leiria, 2411-901 Leiria, Portugal; 3Portuguese Athletics Federation (FPA), 2799-538 Oeiras, Portugal; 4Escola Superior de Desporto de Rio Maior, Instituto Politécnico de Santarém, 2040-413 Rio Maior, Portugal; 5Centro Interdisciplinar de Estudo da Performance Humana (CIPER), Faculdade Motricidade Humana da Universidade de Lisboa, 1495-751 Oeiras, Portugal; 6Sport and Health Department, School of Health and Human Development, Universidad de Évora, 7000-671 Évora, Portugal; 7Comprehensive Health Research Center (CHRC), University of Évora, 7000-671 Évora, Portugal

**Keywords:** IMU, sprint, high-level athletes, anteroposterior acceleration, step time

## Abstract

In this study, we aimed to assess sprinting using a developed instrument encompassing an inertial measurement unit (IMU) in order to analyze athlete performance during the sprint, as well as to determine the number of steps, ground contact time, flight time, and step time using a high-speed camera as a reference. Furthermore, we correlated the acceleration components (XYZ) and acceleration ratio with the performance achieved in each split time obtained using photocells. Six athletes (four males and two females) ran 40 m with the IMU placed on their fifth lumbar vertebra. The accuracy was measured through the mean error (standard deviation), correlation (r), and comparison tests. The device could identify 88% to 98% of the number of steps. The GCT, flight time, and step time had mean error rates of 0.000 (0.012) s, 0.010 (0.011) s, and 0.009 (0.009) s when compared with the high-speed camera, respectively. The step time showed a correlation rate of r = 0.793 (*p* = 0.001) with no statistical differences, being the only parameter with high accuracy. Additionally, we showed probable symmetries, and through linear regression models identified that higher velocities result in the maximum anteroposterior acceleration, mainly over 0–40 m. Our device based on a Wi-Fi connection can determine the step time with accuracy and can show asymmetries, making it essential for coaches and medical teams. A new feature of this study was that the IMUs allowed us to understand that anteroposterior acceleration is associated with the best performance during the 40 m sprint test.

## 1. Introduction

Sprint events, such as the 60 m (indoor), 100 m, and 200 m (outdoor) sprints, require sprint skill characteristics that involve the capacity to produce higher velocities to run a set distance within a lower time [1]. Furthermore, this can require the capacity to generate maximal force and power in the direction of running [2,3]. The velocity profile can be divided into three phases, the acceleration phase, the constant velocity phase, and the deceleration phase [4], and obtaining their mechanical descriptions can improve the athletes’ sporting performance and prevent injuries [5].

There exist several instruments that can quantify sprint performance and help prevent injuries through biomechanical parameters [6,7]. Photocells are usually applied to determine the average velocity, as well as radar systems that allow for a description of the instantaneous velocity or understanding of the force data through inverse dynamics [5]. Although they obtain data in real time, the information is limited and they do not provide essential performance indicators to coaches and athletes, such as the step rate and step length [5,8,9]. Motion capture video-based systems and force plates are generally considered the gold standard in running assessments [10]. They can characterize technical running aspects or verify that high acceleration values occur when high propulsive forces and low braking forces are produced [11,12]. Furthermore, instrumented treadmills are also applied to determine temporal, kinematics, and kinetics parameters during the stance phase. Additionally, they allow the verification of the ratio of forces related to the best performance during the acceleration phases [1,13]. These kinds of instruments can obtain useful information, but they have limitations in biomechanical studies, such as in their technical execution [14], because they have difficulties in terms of the clothing restrictions when placing landmarks [10,15]. Furthermore, they are expensive [6,7,16], are difficult to transport [17], and they are very time-consuming [18].

Inertial measurement units (IMUs) are micro-electromechanical sensor systems (MEMSs) that incorporate an accelerometer, a gyroscope, and a magnetometer [19,20]. They have been applied in sprint testing due to their ease of application and transportation provided by their small size, as well as their ability to acquire validated parameters continuously and systematically and to monitor athletes’ training in real situations [6,7,14,15]. Recently, Young et al. [10] applied the zero-crossing method for running techniques with velocities between 8 km/h (novice) and 16 km/h (expert) compared with Vicon 3D motion tracking data, obtaining good to excellent agreement with pronation, foot strike location, and ground contact time data. Furthermore, the authors identified that with higher velocities, this agreement decreases [10]. Specifically, in the sprint analysis, Kuznetsov [4] presented an IMU to derive information about the step parameters from acceleration and angular velocity data and identified essential information for coaching during daily workouts with a low time consumption. Kenneally-Dabroski et al. [5] intended to validate a method to determine running symmetry, but the results had poor internal validity. Setuain et al. [7] analyzed 20 m sprint results, comparing the IMU with a force plate, and showed appropriate validity and reliability (mean (SD), force plate vs. IMU) results for the velocity (8.61 (0.85) m/s vs. 8.42 (0.69) m/s), force (383 (110) N vs. 391 (103)N), and power (873 (246) W vs. 799 (241) W) measurements and a high correlation with the step time measurements (r = 0.88). Bergamini et al. [8] proved that IMUs are suitable for estimating stance and stride durations during sprint running due to identifying an average absolute difference of 0.005 s and a 95% limit of agreement for this difference of less than 0.025 s. Schmidt et al. [14] purposed an IMU to detect the step parameters in sprinting and correctly identified 96% of the steps and stance durations with a −2.5 (4.8) ms difference when comparing the values with an Optojump Next instrument. Slawinski et al. [18] aimed to compare the maximal total power rates between multiple IMUs with a radar system, identifying non-significant differences and a high correlation rate (r > 0.81). Macadam et al. [19] studied how a thigh-positioned wearable resistance system with 2% body mass affected the 40 m sprint running performance following a five-week training protocol. When they compared the pre- and post-training results, they identified substantially faster times; substantial increases in the theoretical maximum velocity, theoretical horizontal force, maximum power, flight times, and vertical stiffness, and a substantial decrease in the contact times [19]. Blauberger et al. [21] described a method to determine the ground contact time from the IMU data and correctly detected 97.08% of the steps. Regarding the ground contact time, an underestimation rate of 3.55 ms and a root mean square error of 7.97 ms were obtained, classifying smaller errors at the beginning as compared to the end of the sprint test. In summary, studies have applied many kinds of IMUs in sprint analyses to analyze temporal parameters associated with the stance phase, which were in excellent agreement [10,15] with the low errors for ground contact time (GTC) values [4,8,14,15,21,22] when calculating the step numbers with high accuracy (>95%) [14,21], or when analyzing the asymmetry during the sprinting [5]. Additionally, the kinetics parameters during the sprint were validated [7,18] and demonstrated the importance of their application after the hamstring injuries [23]. Furthermore, different IMU positions have been tested [24] in sprinting. Previous authors applied a full body system [18], while some instruments were placed on the ankles [14,21] or foot [22] and others were placed on the back [4,5], specifically at the level of the first lumbar vertebra (L1) [2,8] or at the level of the third and fifth lumbar vertebrae (L3–L5) [7,17,23]. In addition, different locations and surfaces have been used [24] in sprinting studies. Some studies involved running indoors with a treadmill [10,15,22], but tracks (indoor and outdoor) were used when the authors carried out maximal sprints [2,4,5,7,8,14,18,19,20,21,23,25].

Many studies have focused on the determining parameters associated with the number of steps in a sprint using IMUs [4,8,15,17,21,22]. Some of these studies explored angular parameters [19,20] and kinetic parameters and their relations with injuries [7,18,23]. However, Benson et al. [25] suggested that future studies be conducted in real-world environments, and none of the previous studies correlated the acceleration components (XYZ) and acceleration ratio with the performance achieved in each split time using photocells and high-level athletes. In the present study, we aimed to assess sprint performance using a developed instrument encompassing an off-the-shelf inertial measurement unit (IMU) in order to analyze athlete performance during the sprint, as well as to determine the number of steps, ground contact time, flight time, and step time, using a high-speed camera as a reference. Furthermore, we correlated the acceleration components (XYZ) and acceleration ratio with the performance achieved in each split time, as obtained using photocells. We hypothesized that the number of steps and temporal parameters would have lower mean error rates and that similar performance rates would be obtained when comparing the results from the IMU with the high-speed camera. Additionally, we hypothesized that the anteroposterior, vertical, and ratio acceleration results would have a high correlation with each split time obtained using photocells.

## 2. Materials and Methods

### 2.1. Experimental Overview

The experiment conducted in the present study consisted of a session of training monitoring and control, in which we evaluated two to three maximal sprints and determined the number of steps, step time, ground contact time (GTC), flight time, acceleration components (XYZ), and acceleration ratio using the IMU for comparison, correlated with the same parameters used for the photocells as well as a high-speed camera.

### 2.2. Participants

A total of 30 healthy, high-level athletes (15 males and 15 females) with international representation for their countries and without injuries in the past 6 months were invited. Six athletes—four males (one pole vaulter, 5.61 m; three decathletes, 7280 ± 183 points; age, 25.00 ± 3.08; body mass (BM), 77.90 ± 8.69 Kg; height (h), 1.80 ± 0.07 m) and two females (pole vaulters, 4.47 ± 0.04 m; age, 30.00 ± 2.00; BM, 63.23 ± 1.87 Kg; h, 1.70 ± 0.01 m)—accepted and were included in the study. All athletes were informed about the study objectives and signed the protocol approved by the Ethical Council of Universidade de Évora (GD/46951/2019). This declaration was made in accordance with the ethical rights of the Helsinki Declaration.

### 2.3. Experimental Procedures

The athletes were informed about the objectives and performed a warm-up as indicated by their coaches in order to prepare for the training monitoring and control session. The investigator explained the protocol to all athletes. Before initiating the task, the IMU was calibrated. The athlete stood still for about 30 s, in order to obtain the base acceleration, and performed a previous jump to set the initial criteria for both instruments. The athlete began from the three-point start position and ran for 40 m at maximum speed [2,5,8,18]. The sprint was performed on an indoor track with the IMU placed at L5 as a reference for the CoM [7,17,23]. The instrument was fixed on the skin at position L5 using adhesive tape (see Figure 1). Each coach defined the number of sprints performed by their athlete (between two and three 40 m sprints), with ten minutes of rest between runs, performing a total of 14 runs.

### 2.4. Instruments

In this study, we used an instrument developed by the investigation group, which encompasses an off-the-shelf IMU (size, 42 mm × 32 mm × 17 mm; weight, 22 g) consisting of a three-dimensional (3D) accelerometer (±16 g), a 3D gyroscope (±2000 dps), and a 3D magnetometer (±4900 µT), with a sample rate of 300 Hz (see Figure 1). We assembled an ICM20948 (InvenSense, TDK Tokyo, Japan) with an ESP32 (Espressif, Shanghai, China). Our proposal intended to improve some Bluetooth connection problems during the first data collection phase, so we proposed a device with a wireless connection through Wi-Fi with an interface (laptop/tablet). The device was covered with a box of biodegradable polylactic acid (PLA) material designed and printed with an MK3S Prusa 3D Printer (Prague, Czech Republic). A Panasonic Lumix Fz200 high-speed camera (Osaka, Japan) collecting at 100 Hz and four photocells of information, which were double-cut (Bolzano, Italia) as the reference criteria. The IMU data were collected using the Spyder package in Python 3.7. The communication between the IMU and the laptop was conducted using wireless communication (Wi-Fi) in order to improve the connection quality. Figure 2 represents the operating steps of the IMU.

### 2.5. Data Processing

Data processing for the high-speed camera was performed using the Kinovea software which determined the step numbers, GTC, flight phase time, and step time. The photocells obtained direct split times of 0–10 m, 10–20 m, 20–30 m, 30–40 m, and 0–40 m. Their results were organized using Microsoft Excel Office.

The IMU analyses were performed using Scylab 6.0.1 (ESI Group, Paris, France). The anteroposterior (XX), mediolateral (ZZ), and vertical (YY) axes were considered. The acceleration data from the IMU were smoothed using a Butterworth low-pass filter, and the cut-off frequency was determined using a spectral power analysis [26,27]. A cut-off frequency of 10 Hz was applied. The user defined the initial position on the acceleration data and the routine automatically determined the number of steps for each phase (0–10 m, 10–20 m, 20–30 m, 30–40 m, and 0–40 m) through the vertical axis (Figure 3A). The step time was calculated from the amplitude between the maximum peaks. The GTC and flight time phase were determined from the amplitude maximum peak and minimum peak and the minimum peak and maximum peak, respectively (Figure 3B). Furthermore, the acceleration ratio was calculated based on the protocol used by Morin et al. [13], using Equations (1) and (2):
(1)$$R_{Acc}=\sin\theta$$
(2)$$\theta =\tan^{-1}\left(\frac{a_{xx}}{a_{yy}}\right)$$
where $R_{Acc}$ represents the ratio acceleration; θ is the absolute angle between the anteroposterior (horizontal) acceleration ($a_{xx}$) and vertical acceleration ($a_{yy}$). Additionally, the possible differences between the left and right sides through the ground contact time were represented by the best performance over 40 m.

### 2.6. Statistical Analysis

All sprints performed by the athletes were considered in the analysis. The descriptive statistics were calculated to describe the number of steps, GTC, flight time, step time, ratio acceleration, and XYZ acceleration. The normality was verified using the Shapiro–Wilk test (*p* ≤ 0.05) [28]. An independent samples *t*-test was conducted. When the normality was violated, the Mann–Whitney test was used to compare the number of steps, GTC, flight time, and step time results between the IMU and high-speed camera. The accuracy was determined through the mean error, absolute mean error, standard deviation error, and correlation [28,29]. Finally, the linear regression between the IMU acceleration and the average velocity obtained by the photocells was calculated. Pearson and Spearman’s correlations were determined to analyze whether the parameters had the same tendency between instruments. The correlation values suggested by Hopkins et al. [30] were considered: r values ≤ 0.3, small; r values between 0.3 and 0.5, moderate; r values > 0.5, high [5,7]. All statistical analyses were carried out using the Jamovi software (version 1.6; Sydney, Australia).

## 3. Results

### 3.1. Descriptive Analysis

Table 1 provides the results of the descriptive statistical analyses for the analyzed parameters of the IMU and high-speed camera. Regarding the number of steps, the mean number of steps between 0 and 10 m was (mean (SD)) 7.43 (0.756) for the IMU and 7.36 (0.497) for the high-speed camera. The mean number of steps between 10 and 20 m was 5.50 (0.519) for the IMU and 5.29 (0.469) for the high-speed camera. A mean number of steps between 20 and 30 m of 4.71 (0.469) and 5.07 (0.267) were determined for the IMU and high-speed camera, respectively. The number of steps between 30 and 40 m was 4.93 (0.475) for the IMU and 4.36 (0.497) for the high-speed camera. The mean number of steps between 0 and 40 m for the IMU data was 22.6 (0.938), while that for the FP was 22.1 (1.210). About the temporal parameters, the mean ground contact time (mean (SD)) of 0.119 (0.010) s and 0.119 (0.012) s were calculated for the IMU and high-speed camera, respectively. The mean result of flight time for the IMU was 0.118 (0.012) s, while that for the high-speed camera was 0.128 (0.005) s. The mean step time was 0.238 (0.014) s for the IMU and 0.247 (0.011) s for the high-speed camera.

### 3.2. Accuracy Data

Table 2 summarizes the results of the accuracy analysis between the IMU and high-speed camera with respect to the analyzed parameters. For the number of steps over 0–10 m, we identified 93% of the steps performed and obtained a mean error (SD) of −0.071 (0.703) and an absolute mean error of 0.500, with a correlation (r) of 0.374 (*p* = 0.188). When comparing the results between the IMU and high-speed camera, no significant differences were observed (t = 86.0, *p* = 0.555). For the number of steps over 10–20 m, we identified 96% of the steps performed and obtained a mean error (SD) of −0.214 (0.410) and an absolute mean error of 0.214, with a correlation rate of 0.632 (*p* = 0.015). When comparing the results between the IMU and high-speed camera, no significant differences were found (t = 77.0, *p* = 0.266). For the number of steps over 20–30 m, we identified 93% of the steps performed and obtained a mean error (SD) of 0.357 (0.479) and an absolute mean error of 0.357, with a correlation rate of 0.175 (*p* = 0.549). When we compared the results between the IMU and high-speed camera, significant differences were found (t = 65.0, *p* = 0.025). For the number of steps over 30–40 m and 0–40 m, we identified 89% and 98% of the steps performed, respectively, and obtained mean errors (SDs) of −0.571 (0.495) and −0.500 (0.627) and absolute mean errors of 0.571 and 0.500, with correlation rates of 0.438 (*p* = 0.117) and 0.875 (*p* < 0.001), respectively. When comparing the results between the IMU and high-speed camera for the number of steps over 30–40 m, significant differences were observed (t = 46.5, *p* = 0.007), while for the number of steps over 0–40 m, no significant differences were observed t = 73.0, *p* = 0.239). For the GTC, we obtained a mean error (SD) of 0.000 (0.012) s and an absolute mean error of 0.011 s, with a correlation of 0.037 (*p* = 0.904). When comparing the results between the IMU and high-speed camera, no significant differences were found (t = 0.059, *p* = 0.953). For the flight time, we obtained a mean error (SD) of 0.010 (0.011) s and an absolute mean error of 0.012 s, with a correlation of 0.378 (*p* = 0.182). When comparing the results between the IMU and high-speed camera, significant differences were found (t = −2.798, *p =* 0.010). For the step time, we obtained a mean error (SD) of 0.009 (0.009) s and an absolute mean error of 0.012 s, with a correlation rate of 0.793 (*p* = 0.001). When comparing the results between the IMU and high-speed camera, no significant differences were found (t = −1.922, *p* = 0.066).

Figure 4 shows the correlations of the analyzed parameters between the IMU and the high-speed camera for the sprint testing.

### 3.3. Analyzing Symmetries or Asymmetries

Figure 5 shows a comparison of the step time values between the left and right sides.

### 3.4. Linear Regression Models between IMU Acceleration and the Average Velocity Obtained by the Photocells

Table 3 provides the results of the statistical correlation analyses for the acceleration values obtained with the IMU and the average velocity values obtained with the photocells for the distances of 0–10 m, 10–20 m, 20–30 m, 30–40 m, and 0–40 m. For the distance of 0–10 m, the results presented significant correlations. For the distance of 10–20 m, the anteroposterior (XX) maximum acceleration showed a positive significant correlation (r) of 0.565 (*p* = 0.035) and a mean (SD) of 19.00 (2.37) m/s^2^. For the distance of 20–30 m, the anteroposterior (XX) maximum and minimum acceleration showed significant positive (r = 0.823 *p* < 0.001) and negative (r = −0.625 *p* = 0.017) correlations and mean values (SD) of 21.00 (4.07) m/s^2^-and 26.90 (8.14) m/s^2^, respectively. For the distance of 30–40 m, the mediolateral (ZZ) maximum acceleration showed a positive significant correlation (r) of 0.793 (*p* = 0.001) and a mean (SD) of 11.80 (5.69) m/s^2^. Lastly, for the distance 0–40 m, the anteroposterior (XX) maximum acceleration showed a positive significant correlation (r) of 0.670 (*p* = 0.011) and a mean (SD) of 23.20 (4.08) m/s^2^.

Table 4 shows the regression models used to explain the velocity at each distance. The velocity values over 10–20 m are significantly predicted by the maximum anteroposterior acceleration (*p* = 0.015). This model is significant for 40.2% of the velocity values over 10–20 m. The velocity over 20–30 m is predicted by the maximum and minimum anteroposterior acceleration values. This model explains 34.8% of the velocity values over 20 to 30 m. The velocity over 30–40 m is predicted by the mediolateral maximum acceleration (*p* = 0.021). This model explains 37.0% of the velocity values performed over 30–40 m. Lastly, the velocity over 0–40 is predicted by the anteroposterior maximum acceleration (*p* = 0.002). This model explains 57.3% of the velocity values during the 40 m sprint test.

### 3.5. Acceleration Images

Figure 6 represents the vertical acceleration results from the IMU for the best performance of each athlete.

## 4. Discussion

In this study, we assessed the sprint results using a developed instrument encompassing an off-the-shelf inertial measurement unit (IMU) in order to analyze athlete performance during a training sprint session to add more information about the IMU’s applications in a real-world environment [25], determining the number of steps, ground contact time, flight time, and step time, using a high-speed camera as a reference. As many studies have focused on the determining parameters associated with the number of steps during a sprint using IMUs [4,8,15,17,21,22], we also correlated the acceleration components (XYZ) and acceleration ratio with the performance achieved for each split time obtained by the photocells. We hypothesized that the number of steps and temporal parameters would have lower mean error rates and that similar performance levels would be obtained when comparing the results from the IMU with the high-speed camera. Additionally, we hypothesized that the anteroposterior, vertical, and ratio acceleration results would have a high correlation with each split time obtained with the photocells.

### 4.1. Accuracy of the Number of Steps

Our developed system encompassing an off-the-shelf IMU for identifying the number of steps showed high accuracy for the distances of 10–20 m and 0–40 m [28]. Additionally, it determined the numbers of steps with high accuracy over 40 m (98%) and for each 10 m phase (89–96%). Schmidt et al. [14] could identify 95.7% of the steps performed over 20 m, while Blauberger et al. [21] identified 97.08% of the steps performed over 50 m. This information could be relevant to coaches because high-level athletes perform six to seven steps during the first ten meters during a sprint [12]. In addition, the use of a defined distance makes it possible to determine the average length and step frequency and to calculate the average step velocity [31]. 

### 4.2. Accuracy of the Temporal Parameters

The developed IMU showed high accuracy [28] and comparable values for the GCT and non-comparable values for the flight time results [17]. Analyzing our errors, the GCT (0.000 (0.012) s) and flight time (0.010 (0.011) s) values were similar to those in the literature (0.11995 s) [21]. Specifically, Bergamini et al. [8], Blauberger et al. [21], and Lee et al. [15] obtained similar mean errors when measuring the GCT values of 0.025 s, 0.0008 s, and 0.0035 (0.0061) s, respectively. As Lee et al. [15] verified higher correlations to the GTC (r = 0.91) when comparing an IMU with a motion analysis system, we also expected a high correlation to the GTC, but we obtained a small correlation, suggesting a low precision and the need to optimize the methods applied here, particularly in the detection of each phase during the step process.

Regarding the step time, we achieved high accuracy [28], with comparable values and high precision. Our mean error rate for the step time was 0.009 (0.009) s, which was between the ranges identified by Bergamini et al. [8], Blauberger et al. [21], and Lee et al. [15]. Regarding the correlation analysis, the step time obtained here showed a higher correlation. Lee et al. [15] verified higher correlations to the step time (r = 0.91) when comparing an IMU with a motion analysis system. Achieving high accuracy and precision for the step time is an essential result in a sprint analysis due to the athletes developing 95% of their maximum step rate during the acceleration phase [12], and the step rate is equal to $\frac{1}{step\;time}\;$ [31].

### 4.3. Analyzed Symmetries and Asymmetries

In our study, we analyzed the symmetry. Figure 4 shows possible asymmetries during the 40 m sprint testing when analyzing the step time. We verified that the step time was lower when the velocity increased [4], reinforcing that the step rate is an essential parameter to sprint at high levels. This information, showing high accuracy, a small error for the stride time (0.009 (0.009) s), and a high correlation rate (r = 0.793), complements Kenneally-Dabrowski et al.’s results [5], who achieved poor internal validity. This result made possible by the IMU is relevant to coaches and the medical field because asymmetries can be associated with technical issues or biomechanical problems, such as the different force levels between the lower limbs [5].

### 4.4. Linear Regression Models between IMU Acceleration and the Average Velocity Obtained by the Photocells

As much of the literature reports on the motion signal, we added new information about IMUs to associate the anteroposterior, mediolateral, and vertical acceleration values with the average velocity obtained by the photocells over the distances of 0–10 m, 10–20 m, 20–30 m, 30–40 m, and 0–40 m. Through the correlation analysis, our model showed that the higher velocities obtained by photocells indicate a high anteroposterior acceleration over 40 m (0–40 m) and for the partial distances of 10–20 m and 20–30 m. Newton’s second law demonstrates a proportionality directly between the acceleration and force. Our results reinforce the suggestions by Colyer et al. [31] and Rabita et al. [12], who suggested that the anteroposterior forces are represented by an athlete’s best performance during the first 30 m of a sprint run. Our hypothesis was that the acceleration ratio would be a predictive parameter during the first 30 m, but here no distance showed significant correlations. The IMU’s position could be one reason for the results obtained because Morin et al. [1,13] calculated the force ratio from the leg angle. 

### 4.5. Recommendations and Practical Applications

The developed IMU is easy to install and transport, with a cost lower or similar to photocells and the collection and transmission of data, are improved for longer distances, operating through a Wi-Fi connection. In addition, our results reinforce the results obtained by coaches for essential parameters in real time to achieve a high performance, such as for the step time [31]. Furthermore, our work differs by showing that the anteroposterior (horizontal) acceleration had the same tendencies when compared with previous sprinting tests with force plates [12,31], which could be used to improve the information regarding the acceleration and force for high-level athletes and coaches. Lastly, the IMU can be used in injury prevention for performance improvements when sharing the information symmetries with medical professionals and coaches [5].

## 5. Conclusions

In this study, we discussed a small instrument based on an off-the-shelf IMU with a Wi-Fi connection that is easily applicable for training monitoring and control during sprint tests (40 m), making it more systematic and improving the knowledge about IMUs in real environments. We aimed to assess the sprint performance using a developed instrument encompassing an off-the-shelf IMU in order to analyze performance during a sprint test, verifying the high accuracy and precision of the parameter results, namely the number of steps and step time, when compared with a high-speed camera as a reference. However, our automatic methods need improvement in dividing the step time by the GTC and flight time. Additionally, with the step time it was possible to show symmetries, making this information useful to athletes, coaches, sports scientists, or medical teams, complementing other studies about symmetries. Much of the literature reports on the number of steps or the step time in sprinting, so we proposed a comparison of the acceleration components (anteroposterior, mediolateral, and vertical) with the velocity obtained at each distance. This new information allowed us to verify the anteroposterior acceleration as an essential parameter during the first 40 m in high-level athletes, as reported in sprint studies that used force plates or instrumented treadmills. We also hypothesized that the acceleration ratio would be an essential parameter during the sprint test, although the same tendency was not verified, so we suggest that future studies determine the acceleration ratio by placing the IMU on the leg. Future research is needed with more participants, correlating acceleration (IMU) with force (force plates) on an instrumented track.

## Figures and Tables

**Figure 1 sensors-23-01761-f001:**
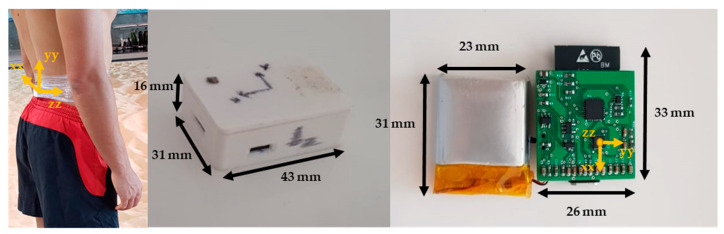
An example of the IMU location at position L5 and the instrument developed by the investigation group.

**Figure 2 sensors-23-01761-f002:**
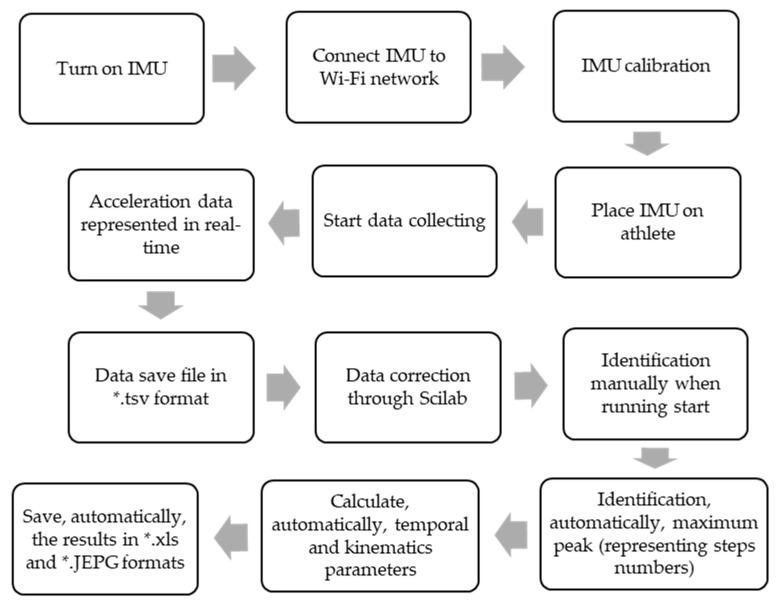
The operating steps of the IMU (for data treatment and processing).

**Figure 3 sensors-23-01761-f003:**
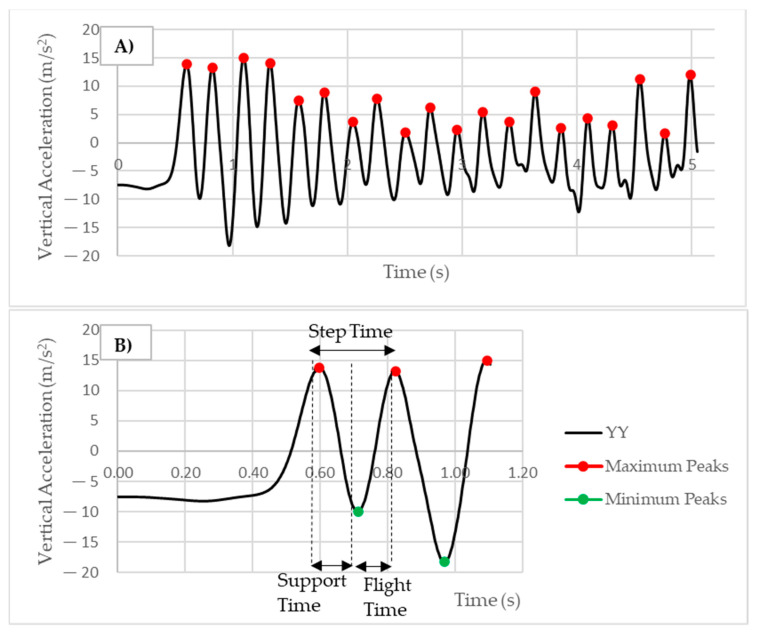
The IMU-obtained vertical acceleration values: (**A**) the identification of the maximum peaks (step numbers); (**B**) the identification of the step time, ground contact time (GCT), and flight time phases.

**Figure 4 sensors-23-01761-f004:**
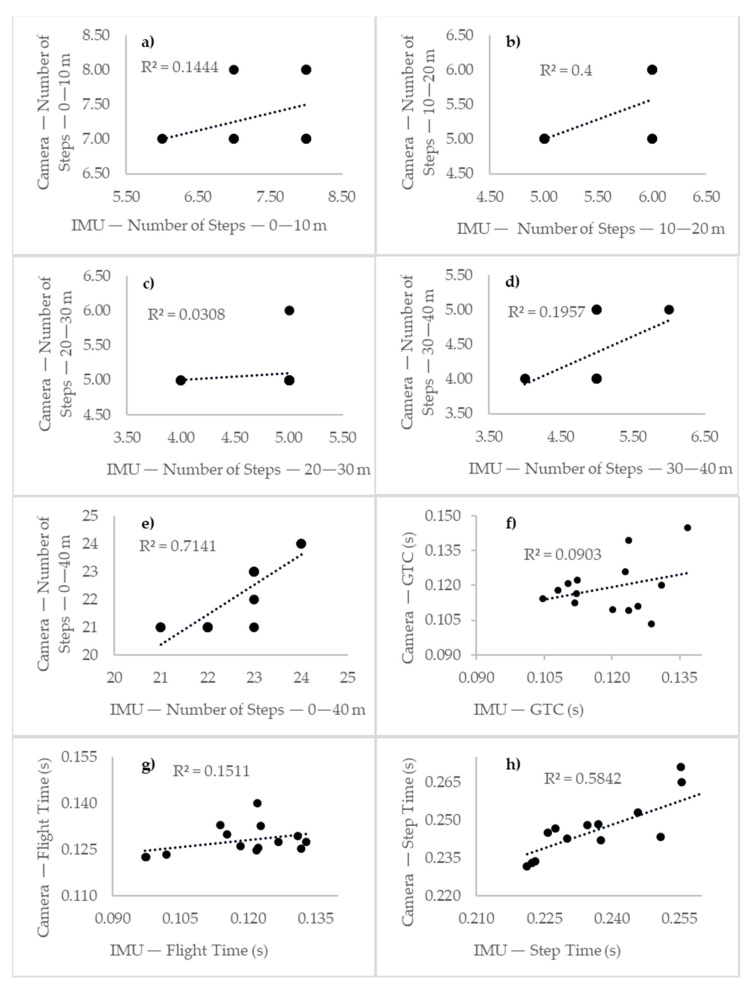
The IMU vs. high-speed camera correlation comparison: (**a**) number of steps—0–10 m; (**b**) number of steps—10–20 m; (**c**) number of steps—20–30 m; (**d**) number of steps—30–40 m; (**e**) number of steps—0–40 m; (**f**) ground contact time (s); (**g**) flight time (s); (**h**) step time (s).

**Figure 5 sensors-23-01761-f005:**
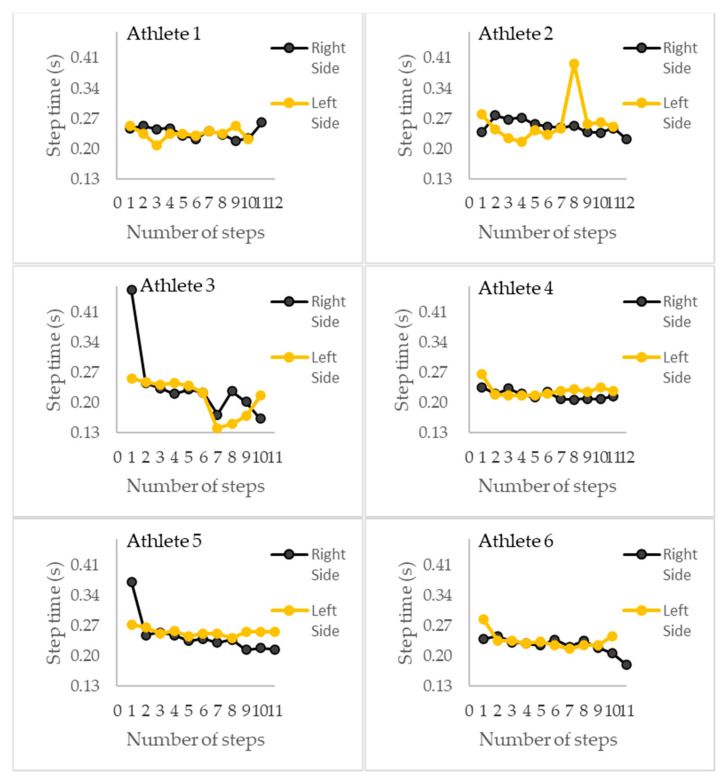
The step time comparison between the left and right sides, showing the best performance results over 40 m for athlete 1, athlete 2, athlete 3, athlete 4, athlete 5, and athlete 6.

**Figure 6 sensors-23-01761-f006:**
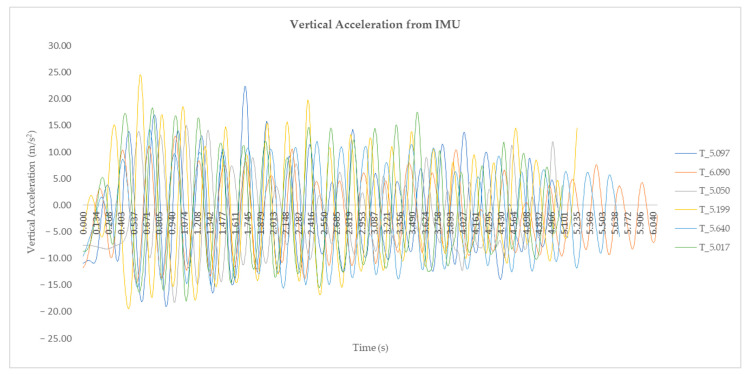
The vertical acceleration results from the IMU for the best performance over 40 m for athlete 1, athlete 2, athlete 3, athlete 4, athlete 5, and athlete 6.

**Table 1 sensors-23-01761-t001:** A descriptive comparison between the inertial measurement unit (IMU) and high-speed camera.

Analyzed Parameter	Mean (SD)	Minimum	Maximum
Number of steps			
0–10 m (IMU)	7.43 (0.756)	6	8 *
0–10 m (Camera)	7.36 (0.497)	7	8 *
10–20 m (IMU)	5.50 (0.519)	5	6 *
10–20 m (Camera)	5.29 (0.469)	5	6 *
20–30 m (IMU)	4.71 (0.469)	4	5 *
20–30 m (Camera)	5.07 (0.267)	5	6 *
30–40 m (IMU)	4.93 (0.475)	4	6 *
30–40 m (Camera)	4.36 (0.497)	4	5 *
0–40 m (IMU)	22.6 (0.938)	21	24 *
0–40 m (Camera)	22.1 (1.210)	21	24 *
Time (s)			
Ground contact time (IMU)	0.119 (0.010)	0.105	0.137
Ground contact time (Camera)	0.119 (0.012)	0.103	0.145
Flight time (IMU)	0.118 (0.012)	0.097	0.133
Flight time (Camera)	0.128 (0.005)	0.123	0.140
Step time (IMU	0.238 (0.014)	0.221	0.261
Step time (Camera)	0.247 (0.011)	0.232	0.271

Note: * non-normal distribution.

**Table 2 sensors-23-01761-t002:** An accuracy comparison between the inertial measurement unit (IMU) and high-speed camera.

Analyzed Parameter	Mean Error (SD)	Abs Mean Error	r	*p*	¥
Number of steps					
0–10 m	−0.071 (0.703)	0.500	0.374	0.188	0.555
10–20 m	−0.214 (0.410)	0.214	0.632	0.015	0.266
20–30 m	0.357 (0.479)	0.357	0.175	0.559	0.025
30–40 m	−0.571 (0.495)	0.571	0.438	0.117	0.007
0–40 m	−0.500 (0.627)	0.500	0.875	<0.001	0.239
Time (s)					
Ground contact time	0.000 (0.012)	0.011	0.037	0.904	0.953
Flight time	0.010 (0.011)	0.012	0.378	0.182	0.010
Step time	0.009 (0.009)	0.012	0.793	0.001	0.066

Note: r, correlation between the IMU and high-speed camera; *p*, *p*-value of the correlation between the IMU and high-speed camera; ¥, *p*-value of the comparisons analysis between the IMU and high-speed camera.

**Table 3 sensors-23-01761-t003:** The description (mean (SD)) and correlation analyses between the inertial measurement unit (IMU) and photocells at a distance of 0–10 m.

			IMU Data—Acceleration (m/s^2^)					
Distance	Photocells (m/s)	Ratio	XX Max	XX Min	YY Max	YY Min	ZZ Max	ZZ Min
0–10 m	5.23 (0.34)	0.52 (0.09)	13.60 (2.31) *	−20.50 (6.12) *	18.70 (5,68)	17.80 (2.31)	13.40 (3.45)	−12.20 (2.99)
10–20 m	8.23 (0.58) *	0.70 (0.11)	19.00 (2.37) ¥	−24.10 (9.37) *	15.50 (5.62)	−13.90 (2.00)	12.00 (4.30)	−11.20 (3.07)
20–30 m	8.97 (0.71)	0.74 (0.11)	21.00 (4.07) ¥	−26.90 (8.14) ¥	12.80 (2.52)	−12.20 (1.61)	12.30 (6.31)	−10.60 (4.69)
30–40 m	9.20 (0.83)	0.74 (0.12)	20.80 (6.61)	−24.90 (6.58)	12.20 (3.11)	−11.40 (1.38)	11.80 (5.69) ¥	−9.27 (4.34)
0–40 m	7.50 (0.54)	0.68 (0.09)	23.20 (4.08) ¥	−29.60 (8.36)	19.70 (5.59)	−17.80 (2.30)	15.00 (5.56)	−13.50 (3.84)

* non-normal distribution; ¥, significant correlation between acceleration obtained by IMU and velocity obtained by photocells.

**Table 4 sensors-23-01761-t004:** The regression coefficients used to explain the acceleration values obtain by the IMU.

	Predictor	β	SE	t	*p*	R	R^2^
Velocity	Intercept	5.289	1.044	5.070	<0.001	0.634	0.402
10–20 m	Maximum acceleration XX	0.155	0.055	2.840	0.015		
Velocity	Intercept	6.807	0.923	7.378	<0.001	0.590	0.348
20–30 m	Maximum acceleration XX	0.076	0.047	1.607	0.136		
	Minimum acceleration XX	−0.022	0.024	−0.926	0.374		
Velocity	Intercept	8.148	0.436	18.690	<0.001	0.609	0.370
30–40 m	Maximum acceleration ZZ	0.089	0.034	2.660	0.021		
Velocity	Intercept	5.197	0.583	8.920	<0.001	0.757	0.573
0–40 m	Maximum acceleration XX	0.099	0.025	4.010	0.002		

## Data Availability

Not applicable.

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
