# Peer review of "Accuracy and Interpretation of the Acceleration from an Inertial Measurement Unit When Applied to the Sprint Performance of Track and Field Athletes"

_sensors, 2023, doi:10.3390/s23041761_

Round 1

Reviewer 1 Report

Summary:

In the article ‘Accuracy and Interpretation of the Acceleration from an Inertial Measurement Units When Applied to the Sprint of Track and Field Athletes’, the authors use an off-the-shelf IMU and apply it to capture the motion of athletes on track. The authors place the IMU on the backside of the athletes, and compute their number of steps, ground contact time and flight time during sprints. The results of the IMU are then compared to the gold standard high-speed camera setup.

Comments for Authors:

1.     Motivation of the research is not clear. The authors mention that the parameters considered in this study (GCT, no. of steps and flight time) have been explored by previous studies. Moreover, studies have also shown that the fifth lumbar location is good for data collection using IMU. Therefore, the authors must justify the novelty of their work and how it is contributing to this field.

2.     What is the off-the-shelf IMU used in this study? Authors should clearly mention the name of the part used for their data collection.

3.     Section 3 (Results) should be improved. The authors have just written the information from the table in words, without providing much additional context. The text in this section is redundant as the tables already contain all the numbers. The authors must improve the text in this section.

4.     The correlation between the data collected from the IMU and the high-speed camera is very poor (Shown in Table 2 and Figure 4). This indicates that the methodology used by the authors is inaccurate, and unable to perform as well as the gold standard. The authors must reassess their data processing algorithm to improve it such that is has high correlation (above 0.7).

Author Response

Dear Reviewer

Thank you for your introduction to our work, commentaries, and suggestions, which were essential to improve our work on IMUs.

Usually, devices with Bluetooth connection are used, but we introduced a device with a Wi-Fi connection. This characteristic improved the connection distance allowing an evaluation in the natural environment to athletes and obtaining real-time and immediate results for the coach.

Regarding motivation (topic 1), we appreciate your commentaries and clarifying the novelty and motivation of the study.

About topic 2, all information requested was clarified in methods.

Regarding topic 3 (results), in the first part, we maintained part of the text because we consider it essential to reinforce the table information. Although, we organized the linear regression models, resuming tables and showing significant statistical results to introduce the presented models.

Thank you for your suggestion 4 about our correlation results. We did not change the results but clarified the precision of the parameters, reinforcing that step time is the only parameter with high accuracy and high precision. We identified that the automatic definition of the GTC and flight time was a limitation of our study.

Lastly, all manuscript was revised by MDPI English Revision.  

One more time, thank you for your commentaries.

With best regards

Reviewer 2 Report

In this work, the authors assessed the sprint using a developed instrument encompassing an off-the-shelf IMU. They correlated the acceleration components and acceleration ratio with the performance achieved in each split time. The results showed that IMU could identify number of steps through each peak. Acceleration obtained by IMU explained sprinting in agreement with the literature. This information may help coaches in the monitoring of the sprint. Some suggestions are as below.

1. The detailed type, specification, and manufacturer of involved materials and devices should be provided.

2. What are the working mechanisms of the device? The authors need to provide working schematics so that more readers will know it better.

3. The manuscript has too many tables. Most of the tables should be presented graphically, which is more intuitive and conducive to analysis.

4. What are the signals of different people running from the IMU vertical acceleration? The authors should provide these data graphs during tests.

5. There is much literature reporting signal detection of motion. What are the different between this work and others, such as materials, methods, the advantages and disadvantages of performance? The authors may compare them with necessary description to better explain the innovation of this work.

Author Response

Dear Reviewer

Thank you for your introduction to our work, commentaries, and suggestions, which were essential to improve our work on IMUs.

Usually, devices with Bluetooth connection are used, but we introduced a device with a Wi-Fi connection. This characteristic improved the connection distance allowing an evaluation in the natural environment to athletes and obtaining real-time and immediate results for the coach.

About topics 1 and 2, we clarify all information requested in the methods section.

Regarding the topic, we reorganized the final part of the results section, resuming tables and information, and showed the essential part, the linear regression models.

In topic 4, we do not understand the question, but we added a section with a Figure with the vertical acceleration of each best athlete's performance.

Lastly, we improve our recommendations and practical application, and conclusion to reinforce our motivation and how our results were important to scientific communities, athletes, coaches, sports scientists, or medical teams.

Additionally, all manuscript was revised by MDPI English Revision.  

One more time, thank you for your commentaries.

Reviewer 3 Report

I have studied the research in detail. I thank the authors for their efforts. The paper is generally well written, the results well presented. However, I have some comments I’d like to express.

The Abstract can be improved by applying the following subheadings: objectives, methods, results, and conclusions

The introduction is too short for readers. This section should be improved, I propose to expand the information on differetn type of sprint and IMUs. More directional literature

The analyzed sample of patients does not support the conclusions in the text, please replace chapter 5 (conclusion):  I highly recommend add the strengths and limitations in this study at the end of the discussion section with a discussion of the methodology used in order to obtain a feasibility study (or proof of concept study)

Author Response

Dear Reviewer

Thank you for your first point about our work. Your commentaries and suggestions were essential to improve our work with IMUs.

Usually, devices with Bluetooth connection are used, but we introduced a device with a Wi-Fi connection. This characteristic improved the connection distance allowing an evaluation in the natural environment to athletes and obtaining real-time and immediate results for the coach.

About topic 2, we tried to improve the abstract with your suggestion’s subheadings.

In topic 3, about motivation, we appreciate your commentaries improving the introduction, specifying studies that apply IMU in sprint tasks.

Regarding topic 4, we improve our recommendations and practical application, and conclusion to reinforce our motivation and how our results were important to scientific communities, athletes, coaches, sports scientists, or medical teams.

Lastly, all manuscript was revised by MDPI English Revision.  

One more time, thank you for your commentaries.

With best regards

Round 2

Reviewer 1 Report

The authors have addressed the all the comments with additional information.

Reviewer 2 Report

The authors have responded to most of my concerns. It can be accepted for publication.